# A Simple Joint Modulation Format Identification and OSNR Monitoring Scheme for IMDD OOFDM Transceivers Using K-Nearest Neighbor Algorithm

**Qianwu Zhang** [ID], **Hai Zhou, Yuntong Jiang, Bingyao Cao, Yingchun Li, Yingxiong Song, Jian Chen \*, Junjie Zhang and Min Wang**

Key Laboratory of Specialty Fiber Optics and Optical Access Networks, Shanghai Institute for Advanced Communication and Data Science, Shanghai University, Shanghai 200072, China; zhangqianwu@shu.edu.cn (Q.Z.); zhouhai3053@shu.edu.cn (H.Z.); jiangtoyun@shu.edu.cn (Y.J.); cby85064@163.com (B.C.); liyingchun@shu.edu.cn (Y.L.); herosf@shu.edu.cn (Y.S.); zjj@staff.shu.edu.cn (J.Z.); wangmin@mail.shu.edu.cn (M.W.)

**\*** Correspondence: chenjian@staff.shu.edu.cn

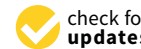

**Featured Application: This paper provides a feasible method for modulation format identification and OSNR monitoring. The application of KNN reduces complexity.**

**Abstract:** In this study, a joint modulation format identification and optical signal-to-noise ratio (OSNR) monitoring algorithm is proposed and experimentally demonstrated using the k-nearest neighbor algorithm for intensity modulation and direct detection (IMDD) orthogonal frequency division multiplexing (OFDM) systems. A modified amplitude histogram of received signal is employed to serve as the classification feature to simplify the computation. Experimental results show that five common quadrature amplitude modulation (QAM) modulation formats, including 4-QAM, 16-QAM, 32-QAM, 64-QAM and 128-QAM, can be identified under 100% accurate estimation at the received optical power of −11 dBm. Robustness of the proposed scheme to constellation rotation is also experimentally assessed. At the same time, system OSNR monitoring also can be achieved and the average prediction mean square error (MSE) is 0.69 dB$^2$, which is similar to that using an artificial neural network. Computational complexity assessment demonstrated that similar performance but less computing resource consumption can be achieved by using the proposed scheme rather than the artificial neural network-based scheme.

**Keywords:** k-nearest neighbor algorithm; modulation format identification; OSNR monitoring

## 1. Introduction

The elastic optical networks [1] (EONs), as one of the solutions of the next-generation fiber-optic communication network, have recently attracted a great deal of attention. Optical orthogonal frequency division multiplexing (OOFDM) is considered a promising alternative to the EONs because of its advantages, including its high spectral efficiency, good resistance to chromatic dispersion, as well as its provision of hybrid dynamic bandwidth allocation in both frequency and time domains. IMDD realizations, e.g., directly modulated laser (DML) based OOFDM transceivers [2], are usually applied in cost-sensitive systems, e.g., optical access networks or data centers, because of their simple structure and DSP (digital signal processing) [3]. With adaptively modulated optical orthogonal frequency division multiplexing (AMOOFDM) [4–9], systems can maximize capacity and achieve flexible bandwidth allocation simultaneously. However, to achieve the bandwidth on demand (BoD) approach, the EONs are expected to be able to dynamically adjust various transmission parameters, e.g., modulation

formats, OSNR, spectrum assignments, etc., depending on the adjusting traffic demands and network condition. Negotiations between the transmitters and receivers are vital for adjusting the modulation format or other parameters before resetting the bit rate. To achieve the flexible bandwidth provision, one of the critical functional requirements for intelligent receivers in EONs is the ability to identify the modulation formats and monitor OSNR of received signals without any prior information through the negotiations between transmitters and receivers.

Modulation format identification has aroused growing attention from researchers in recent years. Bo investigated and characterized a blind modulation format recognition method by projecting partial received data in Stokes space onto a 2-D plane to plot a converted binary graph [10]. Based on the evaluation of the peak-to-average-power ratio of the incoming data samples, Bilal presented a simple novel digital modulation format identification scheme for coherent optical systems after some independent DSP processing at the receiver [11]. Liu proposed a modulation format identification technique based on the extraction and identification of specific features of received signal power distributions in digital coherent receivers, which successfully identified six modulation formats [12]. However, the method described in [10] requires the approximate OSNR and the carrier frequency of the received signals, the method in [11] requires additional hardware components as well as the OSNR of received signals, and the scheme in [12] requires the computation of the received signal power distributions for digital coherent receivers.

In recent years, several machine learning-based modulation format identification (MFI) techniques have been proposed both in digital coherent and directly detected receivers [13–22] for optical communications systems because of their excellent learning ability from data, which can avoid the requirement of pre-information. Khan proposed a deep machine learning method to identify three modulation formats at an accuracy of 100% in a wide optical signal-to-noise ratio range [13]. A simple and cost-effective MFI technique was also proposed by his teams using an artificial neural network based on asynchronous amplitude histogram (AAH) [14]. Guesmi experimentally demonstrated a cost-effective technique to achieve optical performance monitoring functionalities and enable simultaneous symbol rate and modulation format identification based on artificial neural networks [15]. Jiang introduced a novel modulation format identification method based on intensity fluctuation features using support vector machines [16]. Zhang utilized an artificial neural network to identify modulation format and a genetic algorithm to simplify the structures of an artificial neural network for directly detected receivers [17]. However, these methods focus merely on identifying the modulation format of the received signal and do not provide any information about the quality of signal in terms of OSNR. Joint OSNR monitoring and modulation format identification in digital coherent receivers using deep neural network (DNN) was performed in [18]. The excellent experimental effect of DNN was considered, but the problem of high complexity was neglected.

In this study, a joint modulation format identification and OSNR monitoring algorithm is proposed and experimentally demonstrated using the k-nearest neighbor (KNN) algorithm for IMDD OFDM systems. A modified amplitude histogram (AH) of received signal is employed to serve as the classification feature to simplify the computation. Experimental results show that five common QAM modulation formats including 4-QAM, 16-QAM, 32-QAM, 64-QAM and 128-QAM can be identified under 100% accurate estimation at the received optical power of −11 dBm. Robustness of the proposed scheme to constellation rotation is also experimentally assessed. At the same time, system OSNR monitoring can also be achieved with an average prediction MSE of 0.69 $dB^2$, which is similar to what is achieved using an artificial neural network. Computational complexity assessment demonstrated that similar performance but less computing resource consumption can be achieved by using the proposed scheme rather than the artificial neural network (ANN)-based scheme. AMOOFDM without transceiver negotiations can also be achieved using the proposed scheme, showing its good potential for intelligent transceivers in elastic optical networks.

## 2. Operation Principle of Proposed KNN Based Scheme

The KNN algorithm [23] was first proposed by Cover and Hart, which is commonly employed for text categorization, a process of identifying the class to which a text document belongs. The KNN algorithm is considered a simple and intuitive algorithm: comparing the features of a testing data set with a training data set and finding the k instances closest to the instance in the training dataset under a given training dataset, for the new input testing instances. Moreover, the category corresponding to testing data is the one with the largest number of occurrences in k instances. The operation procedure of the KNN algorithm is described as follows:

Firstly, the input dataset is written as:

$$T = \{(x_1,\ y_1),\ (x_2,\ y_2),\ \ldots,\ (x_N,\ y_N)\},\ i = 1,\ 2,\ \ldots,\ N \tag{1}$$

where $x_i \in \chi \subseteq R^n$ denotes the feature vector of the instance; $y_i \in Y = \{c_1,\ c_2,\ \ldots,\ c_k\}$ is the category of the instance. The distance L between input training data and testing data is calculated as Equation (2):

$$L\left(x_i,\ x_j\right) = \left(\sum\nolimits_{l=1}^{2}\left|x_i{}^l - x_j{}^l\right|^2\right)^{\frac{1}{2}} = \sqrt{\left|x_i{}^{(1)} - x_j{}^{(1)}\right|^2 + \left|x_i{}^{(2)} - x_j{}^{(2)}\right|^2} \tag{2}$$

where *i* and *j* denote the instance of training data and testing data, *l* denotes the dimension.

Subsequently, the distance is sorted according to the rising relation of distance.

Next, following the given distance measurement, the k nearest points to *x* are found in the training set T before determining the occurrence frequency of the categories of the preceding *k* points. And the neighborhoods of *x* covering the *k* points are expressed as $N_k(x)$, which is used in Equation (3).

Finally, the category with the highest frequency in the preceding *k* points is returned as the prediction classification of testing data. The majority voting strategy is selected as the decision criteria:

$$y = \frac{\arg\!max}{c_j}\ \sum\nolimits_{x_i \in N_k(x)} I\left(y_i = c_j\right),\ i = 1,\ 2,\ \ldots,\ N; j = 1,\ 2,\ \ldots,\ K \tag{3}$$

where *I* denotes the indicator function. When $y_i = c_j$, *I* is 1, otherwise 0.

As shown in Figure 1, the green dot represents the test data, blue squares represent category A of the training set, and red triangles represent category B of the training set. If the value of *k* is set to 3, the number of red triangles will be 2 greater than the blue squares, of which there is 1. Thus, the category of green dot is red triangle. However, when the value of *k* increases to 5, the category of green dot belongs to blue square since the number of blue squares is greater than the red triangles. The special case of the k-nearest neighbor algorithm is $k = 1$, which is named as the nearest neighbor algorithm.

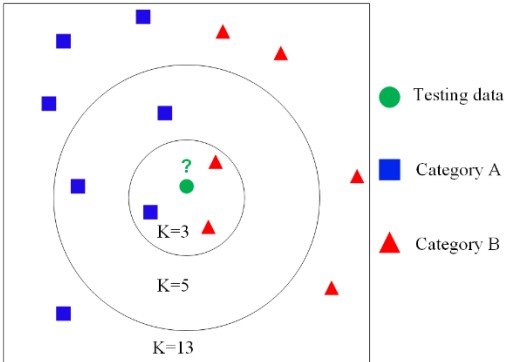

**Figure 1.** The k-nearest neighbor (KNN) schematic diagram.

There are two situations dealing with the complex data. One is the training stage in which modulation format of training data is known, and the other is the testing stage, which is the unknown data waiting for identification with the KNN classifier obtained in the training stage. For the training data, the KNN algorithm is utilized to process the feature vector to build a model which is virtually the KNN classifier. For the unknown data, the feature vector is fed into the KNN classifier, and the modulation format of this data can be obtained. It is worth mentioning that KNN does not have an explicit learning process. In fact, it is a well-known representative of lazy learning. This kind of learning technology only saves samples at the training stage. After receiving the test sample, the subsequent processing is carried out, including comparing the test data with the stored data.

In the receivers of the noted OOFDM systems, fast Fourier transform (FFT) processing is first performed on the received signal to obtain the complex signal data, and then the features of the data can be abstracted from the complex signal data. The AH is introduced in the proposed scheme in which only the real part of the received signals is involved to abstract features. In this study, 30 samples are selected as a training data set, of which every sample is a subcarrier transmission data of OFDM signals. Subsequently, for every subcarrier, 3000 data points are selected from whole 20,400 data points to obtain features. Accordingly, for the 4-QAM, 16-QAM, 32-QAM, 64-QAM and 128-QAM, 2, 4, 6, 8 and 12 peaks should be observed in corresponding AHs. The clustering degree of peaks represents different OSNR. Obviously, when the peak is prominent, the value of OSNR is large.

To simplify the computation complexity and improve the robustness of the algorithm, a data preprocessing scheme is first implemented. The real part of the complex data is divided into 100 intervals from −1.5 to 1.5, after taking real part operations for each data point. The number of points falling in different intervals is used as input feature vector data for KNN. To reduce the effect that computational overhead increases with the increase in data size, each interval is processed as follow:

(a)    If the number of points in an interval ≥256, the value of this interval will be set to 32.
(b)    If the number of points in an interval ≤32, the value of this interval will be set to 0.
(c)    If the number of points in an interval is between 32 and 256, the value of this interval will be set to an integer obtained by using the number divided by 8, which can be conveniently performed by a commercial processor in the future.

Then, the final features of each interval can be obtained.

## 3. Experimental Verification and Discussions

An IMDD OFDM transmission system over an SSMF (standard single mode fiber) as illustrated in Figure 2 is employed to experimentally evaluate the performance of the proposed scheme [24]. Detailed transceiver and system key parameters can be found in Table 1. In this system, all DSP procedures for both transmitter and receiver are achieved by an offline approach. At the transmitter side, the input pseudo random data (PRBS15) is first mapped into parallel complex data with five modulation formats from 4-QAM to 128-QAM. A 64-point inverse fast Fourier transform (IFFT) module is then applied for the generation of OFDM time-domain symbols, in which 30 of them can be used to allocate user data to satisfy the Hermitian symmetry for a real-valued IMDD signal approach. Next, an arbitrary waveform generator (AWG) is employed to generate analog signals at a sampling rate of 2 GSa/s. The analog OFDM signals are then fed into the preamplifiers to adjust the signal amplitude before directly driving a DFB laser at 1550 nm with bandwidth of 2 GHz. Subsequently, the DFB laser converts the electrical signals to optical signals and sends them to the 25 km SSMF.

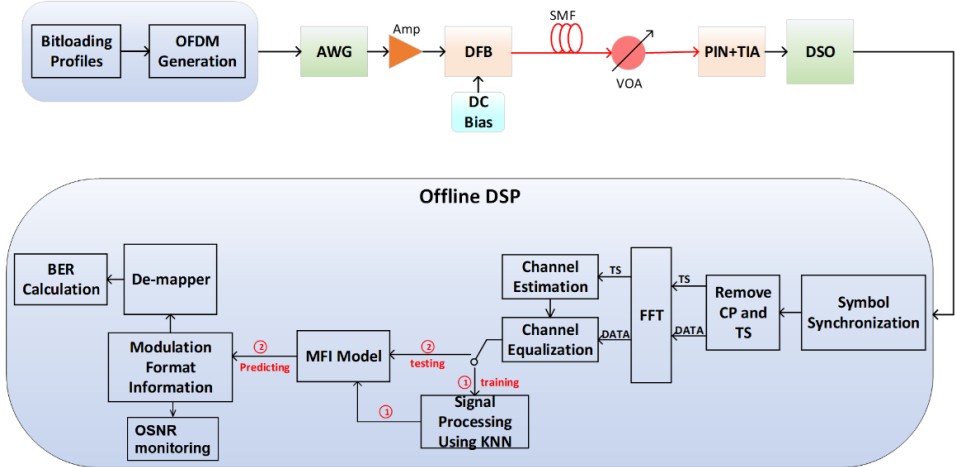

**Figure 2.** The experimental setup of intensity modulation and direct detection (IMDD) orthogonal frequency division multiplexing (OFDM) system.

**Table 1.** Transceiver and system parameters.

| Parameter | Value |
|---|---|
| Modulation format | 128/64/32/16/4-QAM |
| IFFT/FFT size | 64 points |
| CP length | 16 points |
| Bit rate | 4.28 Gb/s |
| OFDM symbols per frame | 100 symbols |
| SSMF length | 25 km |
| DFB modulation bandwidth | 2 GHz |
| DFB wavelength | 1550 nm |
| PIN detector bandwidth | 12 GHz |
| AWG sampling rate | 2 GS/s |
| DSO sampling rate | 10 GS/s |
| DSO/AWG resolution | 8 bit |
| Training set sample size | 30 sample |
| Data points per sample | 3000 points |

At the received side, a 12 GHz PIN (positive intrinsic-negative) with trans-impedance amplifier (TIA) is utilized for O-E conversion by directly detecting the optical OFDM signals. The received optical power (RoP) can be adjusted by a variable optical attenuator (VOA). The received signals are then captured by the digital storage oscilloscope (DSO) with 10 GSa/s sampling rate ADC to convert analog signals to digital signals for offline DSP processing.

During the offline DSP processing, the received data is converted to complex data after symbol synchronization, cyclic prefix (CP)/training sequence (TS) removal, FFT, and channel equalization, as shown in Figure 2. The model of KNN is obtained by training, and then the modulation format is predicted by this model. Once the modulation format is realized using the proposed scheme, the complex data can be decoded to get the original transmit data and to calculate the BER (bit error rate). Meanwhile, OSNR is predicted based on the corresponding modulation formats. The specific recognition and prediction algorithm based on KNN is explained in detail in Section 2.

To investigate the feasibility of the proposed scheme, subcarrier bit loading profiles corresponding to a total bit rate of 4.28 Gb/s shown in Figure 3 are employed for the measurements, including five different mixed QAM modulation formats from 4-QAM to 128-QAM. In this paper, we intentionally chose the bit loading profiles to ensure that every type of modulation format was introduced in the experiment. In fact, in practical application, the proposed algorithm will certainly be used with an adaptive bit-and-power-loading algorithm. The constellations and corresponding abstracted

representative AHs when the RoP is −11 dBm are shown in Figure 4a. The Figure suggests that different peak profiles can be observed for different modulation formats. We can see from Figure 4a that the constellation maps of 4-QAM and 16-QAM have good performance, and that the modulation formats can be easily distinguished. When it comes to 32-QAM, the constellation becomes blurred, but obvious peak value can be seen by the AH method, which means that the modulation format can be well recognized, as shown in Figure 4b. When the modulation formats are 64-QAM and 128-QAM, not only the constellation map is blurred, but also there is no obvious peak value in the AH map. Therefore, KNN is used to process data to get the correct modulation format. As shown in Figure 4b, the shape of AH when RoP is −11 dBm is different from that when RoP is −6 dBm. Thus, as long as the corresponding OSNR under each RoP is known, OSNR value can be predicted according to different AH under different RoP. Consider that KNN has the function of regression prediction [25,26], which can be used in achieving OSNR monitoring [18].

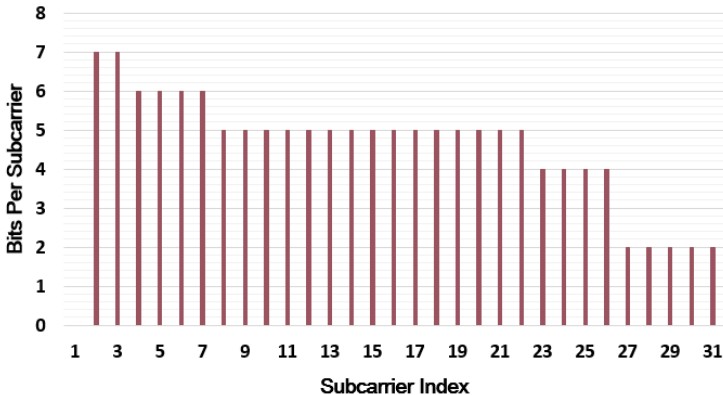

**Figure 3.** Subcarrier bit allocation profile for orthogonal frequency division multiplexing (OFDM) signals.

　　　Since identification accuracy is affected by the number of training samples and the *k* value of the KNN algorithm, two different sets of samples and *k* value, being 30, 1 (case 1) and 60, 3 (case 2) are introduced in the construction of the classifier. The accuracy identification for 4 different sets of bit loading profiles while maintaining the same bit rate under case 1 and case 2 are shown in Figure 5. In general, the accuracy increases with the increase in the number of samples and the *k* value. Besides, there is no exception in this measurement. However, the improvement in accuracy is not significantly obvious as the performance of case 2 (dotted line) is only slightly improved at the RoP of <−12 dBm and cannot reach 100%. Once the RoP reaches −12 dBm or higher, the performance of the two mentioned cases will be the same. This suggests that the number of training samples and the *k* value are not the main influencing factors of the proposed algorithm. To increase the transmission efficiency, the number of training samples is 30, and the *k* value is 1 in the proposed KNN training model during the measurements. As shown in Figure 4, when RoP is −11 dBm, the constellation map is already blurred. However, the identification accuracy of this algorithm can reach 100%, which shows that the algorithm also has a good recognition effect in the case of a relatively high error rate.

　　　Subsequently, the above measurement is repeated under different RoP for both optical back to back (OBTB) and 25 km SSMF configurations. The experimental results involving the mentioned 4 sets of different bit loading profiles are shown in Figure 6, in which the BERs under each bit loading profile is also plotted. Note that the identification accuracy of proposed MFI increases with the increase in RoP. Almost the same performance is obtained for 25 km SSMF transmissions compared to the BTB case, which shows that fiber dispersion does not significantly affect BER and the proposed MFI for current system configuration. This is mainly because our focus in this paper is on the algorithm verification for different types of modulation format, the signal rate involved in our experiment is not very high, and the chromatic dispersion induced system degradation is relatively small. For both 25 km SSMF and OBTB cases, 100% accurate identification can be achieved when the RoP is higher than −11 dBm.

For the RoP >−10 dBm, the BER performance of OBTB and 25 km SSMF can be lower than the adopted HD-FEC limit of $3.8 \times 10^{-3}$ and power penalty is <0.5 dB for 4 different bit loading profiles. System performance with longer distance and higher signal rate will be studied in future works.

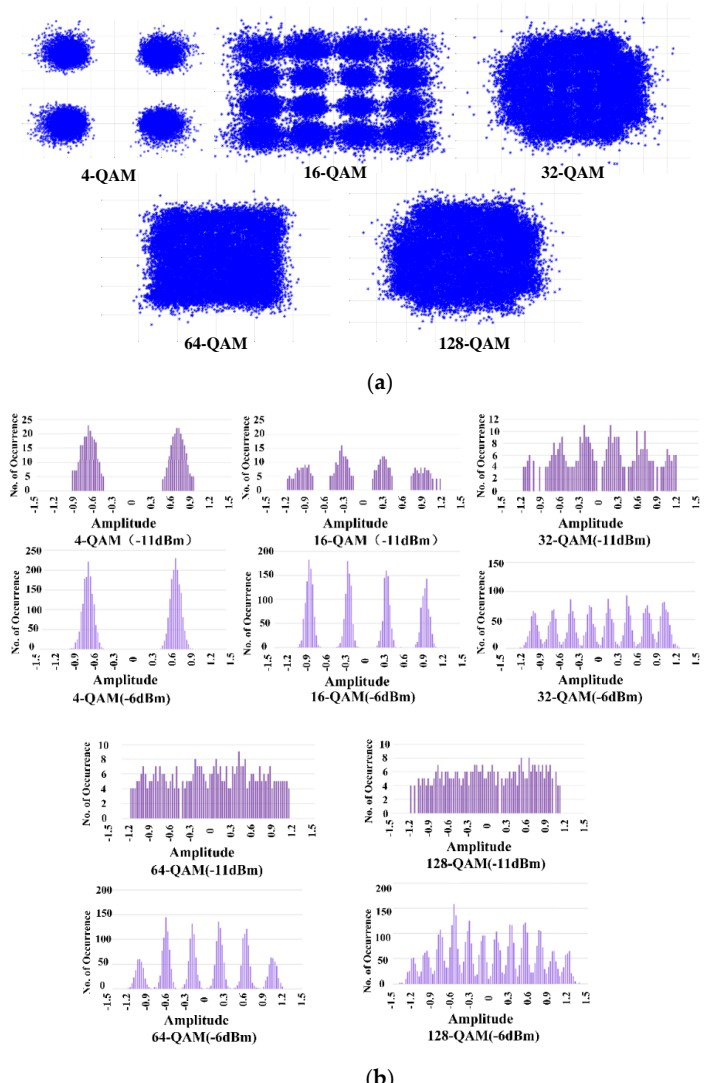

(**a**)

(**b**)

**Figure 4.** (**a**) The constellations, (**b**) AH (RoP = −11 dBm and −6 dBm) of mixed QAM modulation format from 4-QAM to 128-QAM.

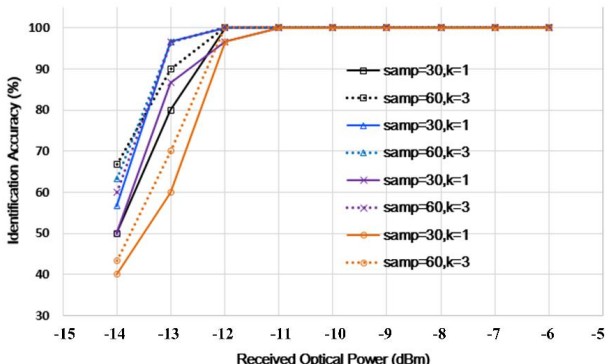

**Figure 5.** The comparison of identification accuracy with different samples and *k* value.

From Figure 6, identification accuracy depends on the receiving optical power, that is, the signal-to-noise ratio. Employment of a stronger equalization algorithm can improve the BER performance and also should improve the identification accuracy at low optical power regions. In practical deployment, the appropriate algorithm can be selected according to the requirements to achieve a balance between performance and complexity.

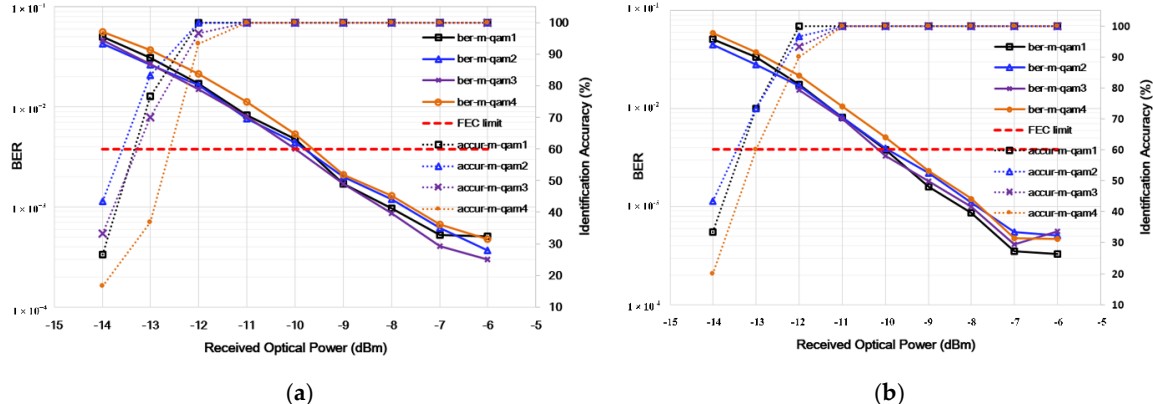

(a)    (b)

**Figure 6.** The BER curves and identification accuracy for (**a**) OBTB and (**b**) 25 km SSMF configurations.

In practical application, constellation rotation after equalization affects the performance of MFI performance. Further experimental investigation is undertaken to assess the effects of constellation rotation. The bit loading as shown in Figure 3 is employed, and additional constellation rotation of $\pi/64$, $\pi/48$, $\pi/32$ and $\pi/28$ is added during the measurements by adjusting the initial phase of the QAM constellation in offline DSP procedures of transmitter. The identification accuracies under different additional phase rotations are shown in Figure 7. It is noteworthy that the identification accuracy increases with the increase in RoP. For the additional phase rotation of $\pi/32$, 100% accuracy cannot be achieved until the RoP meets −6 dBm for the 25 km SSMF case and −7 dBm for the OBTB case. However, for the additional phase rotation of $\pi/28$, 100% accuracy cannot be achieved even under RoP of −6 dBm. For the additional phase rotation of $<\pi/32$, identification accuracy is nearly unchanged compared with the configuration without additional phase rotation, suggesting good robustness to residual constellation rotation.

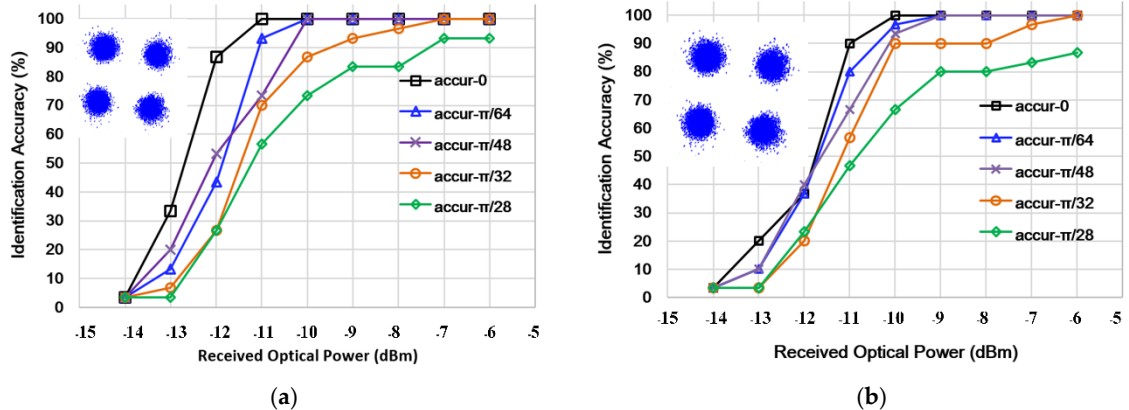

(a)    (b)

**Figure 7.** The identification accuracy under different residual phase rotation for (**a**) OBTB and (**b**) 25 km SSMF configurations. 4-QAM rotated constellation diagram for the additional phase rotation of $\pi/32$ at a specific RoP (−11 dBm) is embedded.

When it comes to OSNR monitoring, the $k$ value should also be determined first. As shown in Figure 8a, as the $k$ value increases, the difference between the estimated and true OSNR increases in general. Specifically, when the $k$ value is small, the estimated OSNR is approximately proportional

to the true OSNR. However, when the *k* value is large, the estimated OSNR is almost unaffected by the true. The MSE between estimated and true OSNR is then calculated to determine the value of *k*. The result is as shown in Figure 8b. When *k* = 2, MSE is the smallest, so *k* is set to 2 when performing OSNR monitoring.

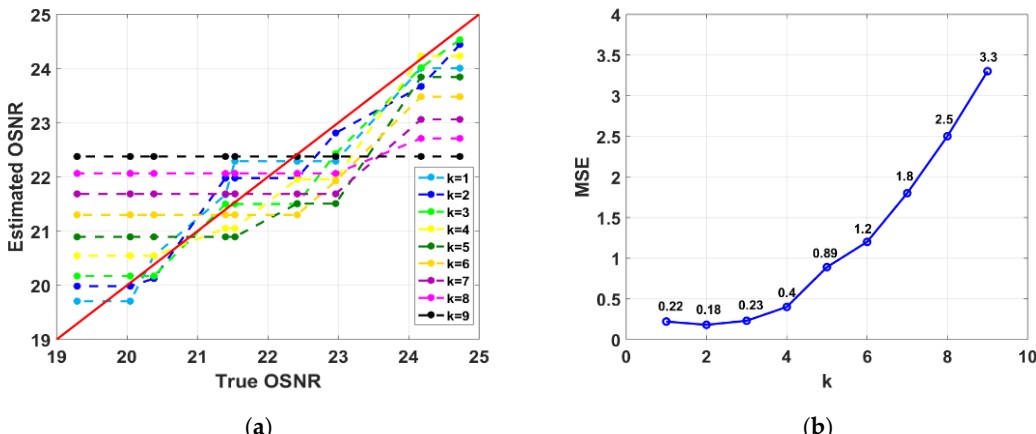

(a)                                     (b)

**Figure 8.** (**a**) Estimated versus true OSNRs with different *k* values and (**b**) MSE versus *k* values for 128 QAM (OBTB).

The OSNR monitoring results for five signal types are shown in Figure 9. It is clear from the figure that OSNR estimates are quite accurate for both OBTB and 25 km SSMF configurations. Hence, the mean OSNR estimation error for the three signal types considered in this work is 0.69 dB, which is similar to the ones reported for the OSNR monitoring [18]. It is worth mentioning that the proposed algorithm can also work for coherent optical systems and single-carrier modulation.

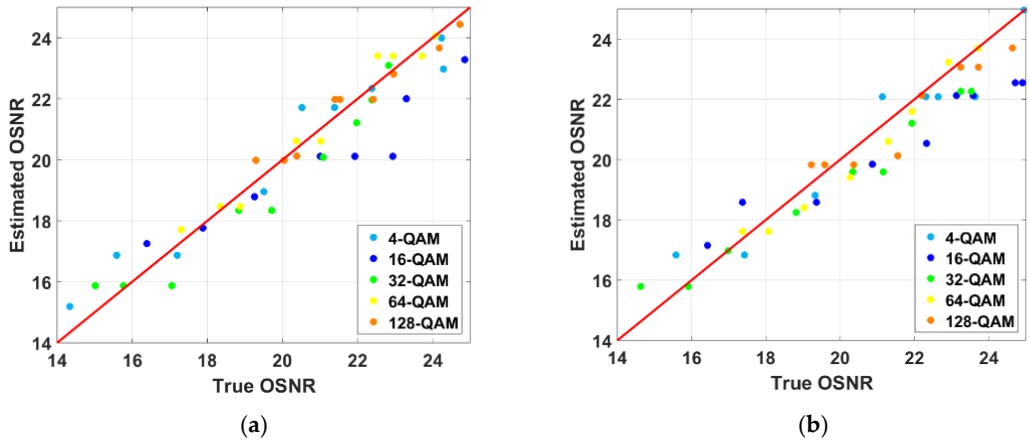

(a)                                     (b)

**Figure 9.** Estimated versus true OSNRs for (**a**) OBTB and (**b**) 25 km SSMF configurations.

We also compared the complexity of the proposed KNN algorithm with artificial neural network [13]. The complexity calculation of ANN [27,28] and KNN are listed in Table 2, in which the complexity calculation involves two parts, namely the training part and the prediction part. For ANN, $N_{ep}$ is the number of samples in a training set and $n_i$, $n_{hid}$ and $n_o$ are the number of neurons on the input, hidden and output layers, respectively. For KNN, $N_{TS}$ is the number of samples in a training set and n is the number of features. The identification accuracy results of both algorithms under the same condition are shown in Figure 10 and the identification accuracy is similar. MSE of OSNR monitoring for both algorithms when RoP is −11 dBm is listed in Table 3. The average MSE of KNN algorithm is 0.69 dB$^2$, and that of the ANN algorithm is 0.71 dB$^2$. In general, compared to the ANN, the KNN

algorithm can effectively reduce multiplication operations. However, the KNN algorithm has similar performance to the ANN algorithms.

**Table 2.** The complexity comparison between KNN and ANN.

| Algorithm | Multiplications | MFI | OSNR Monitoring |
|---|---|---|---|
| ANN | $C_{train} = N_{ep}(n_i n_{hid} + n_{hid} n_o)$ $C_{predict} = n_i n_{hid} + n_{hid} n_o$ $C_{ANN} = C_{train} + C_{predict}$ | $C_{train} = 100 \times (101 \times 4 + 40 \times 5) = 424,000$ $C_{test} = 101 \times 40 + 40 \times 5 = 4240$ $C_{ANN} = 428,240$ | $C_{train} = 100 \times (101 \times 40 + 40) = 408,000$ $C_{test} = 101 \times 40 + 40 = 4080$ $C_{ANN} = 412,080$ |
| KNN | $C_{KNN} = C_{train} + C_{predict} = 2C_{train} = 2N_{TS}n$ | $C_{KNN} = 2 \times 30 \times 101 = 6060$ | $C_{KNN} = 2 \times 30 \times 101 = 6060$ |

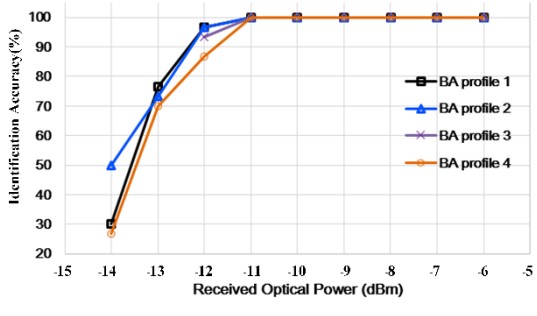

(**a**)

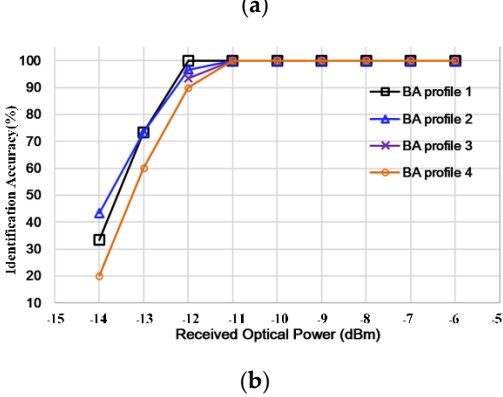

(**b**)

**Figure 10.** The identification accuracy of (**a**) ANN algorithm and (**b**) KNN algorithm.

**Table 3.** The MSE of KNN and ANN algorithm.

| MSE | 4-QAM | 16-QAM | 32-QAM | 64-QAM | 128-QAM |
|---|---|---|---|---|---|
| KNN($dB^2$) | 0.66 | 1.3 | 0.77 | 0.3 | 0.42 |
| ANN($dB^2$) | 0.87 | 0.65 | 0.69 | 0.84 | 0.49 |

## 4. Conclusions

In this study, a joint modulation format identification and OSNR—monitoring algorithm is proposed and experimentally demonstrated using the KNN algorithm for IMDD OFDM systems. A modified AH of received signal is employed to serve as the classification feature to simplify the computation. According to the experimental results, five common QAM modulation formats can be identified with a 100% accurate estimation at the RoP of −11 dBm. At the same time, system OSNR monitoring also can be achieved with an average prediction MSE of 0.69 $dB^2$, which is similar to that using artificial neural network. Robustness and computational complexity of the proposed scheme are also experimentally assessed.

**Author Contributions:** Q.Z. put forward the research and innovation points of the article and revised the paper. H.Z. carried out experimental verification and prepared the paper. Y.J., B.C., Y.L., Y.S., J.C., J.Z. and M.W. help provide funding and experimental environments.

**Funding:** This research was funded by National Natural Science Foundation of China (Project No. 61601279, 61420106011, 61601277, 61635006); Shanghai Science and Technology Development Funds (Project No. 17010500400, 16511104100).

**Conflicts of Interest:** There is no conflict of interest.

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
