# Peer review of "A Simple Joint Modulation Format Identification and OSNR Monitoring Scheme for IMDD OOFDM Transceivers Using K-Nearest Neighbor Algorithm"

_applsci, doi:10.3390/app9183892_

Round 1

Reviewer 1 Report

Dear all,

Enclose you will find my comments.

Best regards

Reviewer 2 Report

Authors propose a joint modulation format identification and OSNR monitoring algorithm, supported by experiments using KNN and a modified amplitude histogram (AH) for IMDD OOFDM showing that 4-QAM and up to 128-QAM can be 100% recognized at -11 dBm ROP. OSNR monitoring also can be achieved with prediction similar to ANN but with lower complexity. AMOOFDM without transceiver negotiations can also be achieved using KNN-AH. This is a very good application of KNN with AH for modulation format identification and OSNR monitoring and I suggest publication of the article pending minor corrections as listed below:

Line 37: The sentence “…receivers are vital for adjust the modulation format…”, should change to “…receivers are vital for adjusting the modulation format…”. Line 42: Authors mention about frequency offset compensation (CFO) but authors use IMDD OOFDM that doesn’t need CFO. Line 65: The sentence “fluctuation features with support vector machine” should be re-written as “fluctuation features using support vector machines”. Line 73: The sentence “…using K-nearest neighbor (KNN) algorithm...” should be “…using the K-nearest neighbor (KNN) algorithm…” Line 79: MSE is not first defined here. Line 86: The sentence “KNN algorithm [20] proposed by Cover and Hart…” should be re-written as “The KNN algorithm [20] was first proposed by Cover and Hart…” Lines 139-143: The whole algorithm should be written in steps form. Line 158: RoP should be defined I was wondering if an adaptive bit-and-power-loading algorithm can be applied here. Line 185: It is mentioned that “KNN has the function of regression prediction, so it has the potential to achieve OSNR monitoring.” How is this justified? Does KNN performs both regression and classification? Figures have very small captions. Please enlarge them. Line 213: It is indicated that for 25 km SSMF transmission compare to the BTB case fiber dispersion does not affect much the proposed MFI. Is this statement also valid for longer distances? Would it be better to use a nonlinear equalizer algorithm or to fix a potential residual rotation of the constellation diagram (using perhaps another algorithm) to assist in the performance of the MFI algorithm? Line 263: The following sentence must be re-written to be grammatically correct “In general, the KNN algorithm can reduce considerable multiplications than ANN.” The Table seems to have very large captions. Can the proposed algorithm work for coherent optical systems and single-carrier modulation? I would expect more references related to AMOOFDM for IMDD and also for IMDD Fast-OFDM.

Round 2

Reviewer 1 Report

Thanks for the revision. The previous proposal are considered in the new version. The paper is now more clear to me.